# R2D2-Net: Shrinking Bayesian Neural Networks via R2D2 Prior

## Abstract

Bayesian neural networks (BNNs) treat neural network weights as random variables, which aim to provide posterior uncertainty estimates and avoid overfitting by performing inference on the posterior weights. However, the selection of the appropriate prior distributions remains a challenging task, and BNNs may suffer from catastrophic inflated variance or poor predictive performance when poor choices are made. Previous BNN designs apply different priors to weights, but the behaviours of these priors make it difficult to sufficiently shrink noisy signals or easily overshrink important signals in the weights. To alleviate this problem, we propose a novel R2D2-Net, which imposes the $R^2$-induced Dirichlet Decomposition (R2D2) prior to the BNN weights. R2D2-Net can effectively shrink irrelevant coefficients towards zero, while preventing key features from over-shrinkage. To more accurately approximate the posterior distribution of weights, we further propose a variational Gibbs inference algorithm that combines the Gibbs updating procedure and gradient-based optimization. We also analyze the ELBO and develop analytical forms of the KL divergences of the shrinkage parameters. Empirical studies on image classification and uncertainty estimation tasks demonstrate that our proposed method outperforms the existing BNN designs with different priors, which indicates that the R2D2-Net can select more relevant variables for predictive tasks. On the other hand, we empirically show that the R2D2-Net yields relatively better predictive performance and smaller variance with the increase in neural network depth, which indicates that the R2D2-Net alleviates the catastrophic inflation of variance when BNNs are scaled. Codes are anonymously available at `https://anonymous.4open.science/r/r2d2bnn-EF7D`.

## 1 Introduction

In the past decades, deep neural networks (DNNs) have shown great success in solving tasks with high-dimensional features. Most of the state-of-the-art (SOTA) deep neural network architectures adopt frequentist approaches which train a single set of weights. These models cannot address the epistemic (i.e., model-wise) uncertainties, which may cause overfitting with small datasets. Failure to address the model-wise uncertainties would lead to poor inference performance for out-of-distribution data. Such frequentist methods also lack uncertainty estimates as they typically only provide point estimates (Kendall and Gal, 2017). The recent emergence of Bayesian deep learning frameworks provides a practical solution to quantify uncertainties in deep learning models.

Bayesian neural networks (BNNs) refine SOTA deep learning architectures with Bayesian approaches, which enable neural networks to capture uncertainties stemming from models (Jospin, 2020; Shridhar et al., 2019). BNNs also act as a natural regularization technique that mitigates the bias of the model by performing inference based on a pool of posterior model weights. Existing BNN architectures widely adopt zero-mean multivariate Gaussian distributions as the prior distributions for the weights (Shridhar et al., 2019). However, simply assuming a multivariate Gaussian distribution often leads to many unnecessary nodes with large variances. This further results in large variances in posterior predictions. The consequence would be catastrophic because most of the deep BNNs without a proper prior underfit and thus predict randomly. As a consequence, variable shrinkage priors are needed to reduce the noise in coefficients and alleviate the variance inflation issue.

Recently, several works (Ghosh et al., 2019; Popkes et al., 2019; Matsubara et al., 2020; Tran et al., 2022) attempt to adopt global–local shrinkage priors to mitigate the problem of large variance.

These priors are able to shrink the coefficients and alleviate the under-fitting problem in BNNs. Although existing shrinkage priors demonstrate superior performance in variable selection, the properties of these priors are subject to several limitations. For instance, these priors have either a low concentration rate around zero or a light tail. A low concentration rate around zero leads to weak shrinkage effects, while the variance of prediction remains large. A light tail under-weighs the effects of large coefficients, which over-shrinks the important signals (Zhang et al., 2020). In particular, the Gaussian distribution has the lightest tail and assigns almost zero weight to large signals. This leads to over-regularization as well as poor feature representation learning performance, especially when the architecture is deep.

The $R^2$-induced Dirichlet Decomposition (R2D2) prior possesses the largest concentration rate at zero and the heaviest tail (Zhang et al., 2020). This property is crucial to predictive models with a large number of parameters — especially to neural networks. We hence propose a novel BNN design with the R2D2 prior on the neural network weights — the R2D2-Net. The R2D2-Net is more effective in model selection than designs based on other existing shrinkage priors because it can choose more powerful weights in predictive tasks.

**Contribution summary:** (1) We propose a novel BNN design — the R2D2-Net, which improves the shrinkage effect and the predictive performance over existing priors by specifying an R2D2 prior on the model weights. (2) We propose a variational Gibbs sampling algorithm that integrates the Gibbs sampling procedure and gradient-based optimization. It provides a more accurate and robust approximation than conventional variational inference methods. (3) We analyze the ELBO in the variational inference of BNN and develop analytical forms of the KL divergences of the shrinkage parameters. (4) Extensive synthetic and real data experiments validate the performance of R2D2-Net on both predictive tasks and uncertainty estimation tasks compared with a variety of existing BNN designs.

## 2 RELATED WORKS

**Global–Local Shrinkage Priors.** High-dimensional regression often suffers from the curse of dimensionality. This motivates novel approaches to the shrinkage of coefficients and variable selection. Global–local shrinkage priors are a class of shrinkage priors that can be essentially expressed as a global–local scale Gaussian family. Existing shrinkage priors exhibit desirable theoretical and empirical properties that can effectively perform coefficient shrinkage. Carvalho et al. proposed the Horseshoe prior, which exhibits Cauchy-like flat and heavy tails and maintains a high concentration rate at zero. Although the Horseshoe prior and its variants Bhadra et al. (2017); Piironen and Vehtari (2017) present satisfactory properties in shrinking the coefficients, their tail thickness and concentration rates at zero are less desirable compared to some recently proposed global–local shrinkage priors. A higher concentration rate at zero allows the model to shrink unnecessary coefficients toward zero more aggressively, and a heavier tail can avoid shrinking key coefficients that have large values (i.e., strong signals). Zhang et al. proposed the $R^2$-induced Dirichlet Decomposition (R2D2) prior, which specifies a prior based on the $R^2$ of the model fit. This prior demonstrates optimal behaviors both in the tails and at zero, which potentially provides the best shrinkage performance while preserving the important signals in the weights.

**Bayesian Neural Networks.** BNNs specify a prior distribution on the weights and bias parameters of the neural network. A vanilla BNN assumes a zero-mean multivariate Gaussian distribution on the weights. The MC Dropout approach (Gal and Ghahramani, 2016) randomly drops out weights to produce posterior samples from a trained frequentist neural network. Moreover, the variance inflates as the number of layers increases, making deep BNNs extremely difficult to build and optimize (Dusenberry et al., 2020). Most of the existing works focus on smaller architectures (e.g., LeNet) and datasets (e.g., CIFAR 100) while a few works (Lakshminarayanan et al., 2017; Dusenberry et al., 2020) scale up their methods to more modern architectures (e.g., ResNet101).

To address this problem, the aforementioned sparsification methods are adopted to shrink unnecessary neurons to prevent variance inflation. Using sparsity–induced priors (Louizos et al., 2017) has become a more popular approach than variational dropout methods (Molchanov et al., 2017; Smith and Gal, 2018). Ghosh et al. proposed to place the Horseshoe prior on the variance of weights to resolve the large prediction variance problem. However, due to the relatively low concentration rate around zero, the shrinkage effect is limited at scale. At the same time, the relative light tail of Horseshoe than R2D2 limits its capability to preserve important signals, which likely leads to over-shrinkage.

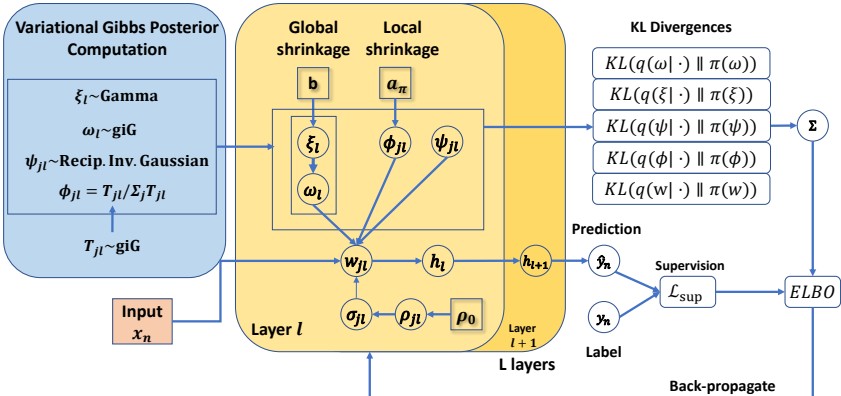

Figure 1: Overview of the proposed R2D2-Net with the yellow part representing the graphical model of each neuron and the blue part summarizing the variational Gibbs inference for computing the posterior distribution of weights.

**Variational Inference of BNN.** Variational inference is a common technique to train the BNNs. Classical BNN training paradigms widely adopt a 'mean-field' approach to approximate the posteriors (which assumes independent marginal distributions) (Ghosh et al., 2019; Shridhar et al., 2019; Molchanov et al., 2017; Rudner et al., 2022; Farquhar et al., 2020). In variational inference, a posterior distribution $p(\boldsymbol{\theta}|\boldsymbol{y})$ is approximated by a variational posterior distribution $q$ selected from a candidate set $\mathcal{Q}$ by maximizing an evidence lower bound (ELBO):

$$\max_{q \in \mathcal{Q}} \mathbb{E}_{\boldsymbol{\theta} \sim q}[\ln p(\boldsymbol{y}|\boldsymbol{\theta})] - \mathrm{KL}(q\|\pi), \tag{1}$$

where $\mathrm{KL}(q\|\pi) = \mathbb{E}_{q \in \mathcal{Q}}[\ln p(\boldsymbol{\theta}|\cdot)] + \mathbb{H}[\pi(\boldsymbol{\theta})]$, is the Kullback–Leibler divergence between $q$ and the prior distribution $\pi$, $\mathbb{H}(\pi(\boldsymbol{\theta}))$ is the entropy of the distribution, and $p(\boldsymbol{y}|\boldsymbol{\theta})$ is the likelihood. Most works (Ghosh et al., 2019; Shridhar et al., 2019; Gal and Ghahramani, 2016; Farquhar et al., 2020) assume Gaussian distributions on weights and hence the Kullback-Leibler (KL) is approximated by $\mathrm{KL}(q\|\pi) = \sum_{j,l} \mathrm{KL}(q(w_{jl}|\cdot)\|\pi(w_{jl}|\cdot))$, where $q(w_{jl}|\cdot)$ is the variational posterior and $\pi(w_{jl}|\cdot) = \mathcal{N}(w_{jl}|0, 1)$ is the standard normal prior on weights. However, the Gaussian assumption is strong and the estimation of KL suffers from large approximation error. By comparing the KL divergence of the analytical distributions of the hierarchical prior (e.g., Horseshoe, R2D2), a more accurate approximation of the ELBO would be obtained.

**Sparsifying Neural Networks.** Another related field of our work is neural sparsification which focuses on "compressing" neural networks to prune unnecessary neurons and improve the space efficiency (Louizos et al., 2017; Molchanov et al., 2017; Han et al., 2016; Srinivas and Babu, 2016). Sparsity-induced prior is also a popular choice in this field (Ghosh et al., 2019; Louizos et al., 2017). Despite the similarity in approaches, our work focuses on a design on BNN with shrinkage priors which can improve its capability (i.e., predictive and uncertainty estimation performance) instead of compressing the existing architecture.

## 3 PRELIMINARIES

**Deep Neural Network (DNN).** A DNN with $L$ layers can be defined as

$$f_l(\boldsymbol{x}) = \frac{1}{\sqrt{D_{l-1}}} \left( W_l \phi(f_{l-1}(\boldsymbol{x})) \right) + b_l, \quad l \in \{1, \ldots, L\},$$

where $\phi$ is a nonlinearity activation function, e.g., the rectified-linear function, $\phi(a) = \max(0, a)$, $D_{l-1}$ is the dimension of input, $b_l \in \mathbb{R}^{D_l}$ is a vector containing the bias parameters for layer $l$, and $W_l$ is the weight tensor. For linear layers, $W_l \in \mathbb{R}^{D_l \times D_{l-1}}$, and for convolutional layers, $W_l \in \mathbb{R}^{D_l \times D_{l-1} \times d_k \times d_k}$, where $d_k$ is the kernel size. Let $\boldsymbol{w}_l = \{W_l, b_l\}$ denote the union of weight and bias parameters of layer $l$, and let $w_{jl}$ denote the $j$-th element of the parameter vector at layer $l$, and let $p_l = |\boldsymbol{w}_l|$. The trainable network parameters are denoted as $\boldsymbol{\theta} = \{\boldsymbol{w}_l\}_{l=1}^{L}$.

**Bayesian neural network (BNN).** The BNN specifies a prior $\pi(\boldsymbol{\theta})$ on the trainable weights $\boldsymbol{\theta}$. Given the dataset $\mathcal{D} = \{\boldsymbol{x}_i, y_i\}_{i=1}^{N}$ of $N$ pairs of observations and responses, we aim to estimate the

posterior distribution of the weights, $p(\boldsymbol{\theta}|\mathcal{D}) = \dfrac{\pi(\boldsymbol{\theta}) \prod_{i=1}^{N} p(y_i|f(\boldsymbol{\theta}, \boldsymbol{x}_i))}{p(\mathcal{D})}$, where $p(y_i|f(\boldsymbol{\theta}, \boldsymbol{x}_i))$ is the likelihood function and $p(\mathcal{D})$ is the normalization term.

## 4 METHODOLOGY

To obtain the best variable shrinkage performance, we impose the R2D2 prior on neural network weights, leading to the R2D2-Net. By placing the R2D2 prior on the weights, the irrelevant weights can be shrunk heavily and the significant weights can be preserved. We also propose variational Gibbs inference and develop analytical forms of KL divergences of the shrinkage parameters to obtain a better estimate of the posterior distribution of weights.

### 4.1 THE R2D2-NET

$R^2$**-induced Dirichlet Decomposition Shrinkage Prior.** Consider a linear model, $Y_i = \boldsymbol{x}_i^\top \boldsymbol{\beta} + \epsilon_i, i = 1, \ldots, N$, where $Y_i$ is the response, $\boldsymbol{x}_i$ is the $p$-dimensional vector of covariates for the $i$-th observation, $\boldsymbol{\beta} = (\beta_1, \ldots, \beta_p)^\top$ is a vector of coefficients, and $\epsilon_i$ is the error term. The R2D2 prior specifies a prior on the $R^2$ from the model fit. The $R^2$ of linear prediction is given by $R^2(\boldsymbol{\beta}) = \dfrac{\text{var}(\boldsymbol{X}^\top \boldsymbol{\beta})}{\text{var}(\boldsymbol{X}^\top \boldsymbol{\beta}) + \sigma^2}$, where $\boldsymbol{\beta}$ can be viewed as the weight tensor of the convolutional or the linear layer and $\boldsymbol{X} \in \mathbb{R}^{n \times p}$ is the data matrix. By specifying a beta prior on $R^2(\boldsymbol{\beta})$, the marginal R2D2 prior has the form,

$$\beta_j \sim \mathcal{N}(0, \psi_j \phi_j \omega \sigma^2/2), \ \psi_j \sim \text{Exp}(1/2), \ \phi \sim \text{Dir}(a_\pi, \ldots, a_\pi), \ \omega|\xi \sim \text{Ga}(a, \xi), \ \xi \sim \text{Ga}(b, 1), \tag{2}$$

where Exp denotes the exponential distribution, Ga denotes the Gamma distribution, and Dir denotes the Dirichlet distribution. The R2D2 prior has the highest concentration rate at zero and heavier tails than other global–local priors (Zhang et al., 2020). Therefore, it can substantially shrink the covariates that do not have effects on the response to zeros. For coefficients that have large signals (i.e., large norms), the heavy-tail nature of the R2D2 prior is able to avoid over-shrinking these coefficients, thus preserving the ability to extract key features from the input data. To compose the R2D2-Net, we place the R2D2 prior on each $w_{jl}$ in each of the linear layers and convolutional layers.

### 4.2 VARIATIONAL GIBBS INFERENCE FOR OPTIMIZATION

Since we have the marginal R2D2 distribution in Eq. (2), we adopt a mean-field approach to the ELBO (i.e., factorize $q(\boldsymbol{\theta})$ into the product of the marginal distribution of each neuron). First, we update $\boldsymbol{w}$ and $\boldsymbol{\rho}$ by back-propagating the ELBO in Eq. (1). We initialize the weight parameters $\boldsymbol{w}_l$ with a reparameterized Gaussian distribution, $w_{jl} \sim \mathcal{N}(\mu_{jl}, \sigma_{jl}^2 \psi_{jl} \phi_{jl} \omega_l)$, where each standard deviation $\sigma_{jl}$ is reparameterized by introducing a parameter $\rho_{jl}$ such that $\sigma_{jl} = \log(1 + e^{\rho_{jl}})$. We assign an individual variance term $\sigma_{jl}$ to each weight, which is different from Zhang et al. who assume the same $\boldsymbol{\sigma}_l = \sigma_l \mathbf{1}$ for all weight parameters in layer $l$. Since the distribution of $\sigma_l$ in Zhang et al. is updated by the regression MSE, which is analogous to learning the variance of neurons by backpropagation of task-specific loss. Therefore, we distinctively set a for each neuron and learn them from backpropagating the task-specific loss to update the variance term $\boldsymbol{\sigma}_l$ in a deep learning setting. We set the prior values of $\boldsymbol{\psi} = \{\psi_{jl}\}_{j=1}^{p_l}{}_{l=1}^{L}, \boldsymbol{\phi} = \{\phi_{jl}\}_{j=1}^{p_l}{}_{l=1}^{L}, \boldsymbol{\omega} = \{\omega_l\}_{l=1}^{L}, \boldsymbol{\xi} = \{\xi_l\}_{l=1}^{L}$ with the prior distribution defined in Eq. (2) and $\mu_{jl} = 0, \rho_{jl} = \rho_0$ for the first step.

With the weight parameter samples, we are able to compute the ELBO using Eq. (1). The trainable parameters $\boldsymbol{w}$ and $\boldsymbol{\rho}$ can be updated by back-propagating the ELBO. We then update shrinkage parameters using the updated $\boldsymbol{w}$ and $\boldsymbol{\sigma}$. Following the Gibbs sampling procedures proposed by Zhang et al., we propose our variational Gibbs inference algorithm to update the shrinkage parameters alternatively using their individual posterior distributions. We first sample $\psi_{jl}, \omega_l$ and $\xi_l$,

$$\psi_{jl}^{-1} \mid w_{jl}, \phi_{jl}, \sigma_{jl}^2 \sim \text{InvGaussian}(\mu = \sqrt{\sigma_{jl}^2 \phi_{jl} \omega_l/2}/|w_{jl}|, \lambda = 1),$$

$$\omega_l \mid \boldsymbol{w}_l, \boldsymbol{\phi}_l, \xi_l, \boldsymbol{\sigma}_l^2 \sim \text{giG}(\chi = \sum_{j=1}^{p_l} 2w_{jl}^2/(\sigma_{jl}^2 \phi_{jl} \psi_{jl}), \rho = 2\xi_l, \lambda_0 = a_l - \frac{p_l}{2}),$$

$$\xi_l \mid \omega_l \sim \text{Ga}(a_l + b_l, 1 + \omega_l).$$

To sample $\phi_l \mid \boldsymbol{w}_l, \psi_l, \xi_l, \sigma_l^2$, we first draw $T_{1l}, \ldots, T_{p_l l}$ independently with $T_{jl} \sim \mathrm{giG}(\chi = 2w_{jl}^2/(\sigma_{jl}^2 \psi_{jl}), \rho = 2\xi_l, \lambda_0 = a_l - \frac{p_l}{2})$, and then set $\phi_{jl} = \frac{T_{jl}}{T_l}$ with $T_l = \sum_j T_{jl}$. We repeat the above steps to train the R2D2-Net iteratively till convergence or early stopping criteria are met (e.g., loss is not improving). The algorithm in the appendix presents the detailed workflow of the variational Gibbs inference, which leverages the advantages of both posterior computation and gradient-based estimation to obtain better approximations of the shrinkage parameters.

## 4.3 Estimation of KL Divergences with Variational Posterior Distributions

In light of the importance of obtaining an accurate estimate of the KL loss in variational inference, we utilize the full posterior distribution obtained in variational Gibbs inference and the R2D2 prior to obtain a more accurate estimate of the KL loss. The KL divergence of the variational posterior $q$ and the prior $\pi$ can be divided into several components:

$$\mathrm{KL}(q(\boldsymbol{\theta}|\cdot)\|\pi(\boldsymbol{\theta})) = \mathrm{KL}(q(\xi|\cdot)\|\pi(\xi)) + \mathrm{KL}(q(\boldsymbol{\omega}|\cdot)\|\pi(\boldsymbol{\omega})) + \mathrm{KL}(q(\boldsymbol{\psi}|\cdot)\|\pi(\boldsymbol{\psi})) \\ + \mathrm{KL}(q(\boldsymbol{\phi}|\cdot)\|\pi(\boldsymbol{\phi})) + \mathrm{KL}(q(\boldsymbol{w}|\cdot)\|\pi(\boldsymbol{w})).$$

We can obtain the closed-form solutions of the KL divergences for $\boldsymbol{\omega}, \boldsymbol{\xi}$, and $\boldsymbol{\psi}$. We approximate the KL divergence of $\boldsymbol{\phi}$ using samples from the variational posterior distribution $q(\boldsymbol{\phi}|\cdot)$. Table 1 presents the closed forms of the KL divergences on the shrinkage parameters $\xi_l, \omega_l$, and $\psi_{jl}$. The closed forms in Table 1 can be obtained by using $\mathbb{E}(X), \mathbb{E}(X^{-1})$ and $\mathbb{E}(\ln X)$ for $X \sim \mathrm{giG}$, which are given in the supplementary materials together with the detailed derivations of the KL losses.

Table 1: Analytical forms of KL-divergences of the shrinkage parameters ($\xi_l, \omega_l, \psi_{jl}$)

| | Prior $\pi$ | Variational Posterior $q$ | Closed Form of KL-divergence |
|---|---|---|---|
| $\xi_l$ | Gamma | Gamma | $\mathbb{E}_q\left[\ln\left(\frac{(1+\omega_l)^{a_l+b_l}}{\Gamma(a_l+b_l)}\xi_l^{a_l+b_l-1}e^{-(1+\omega_l)\xi_l}\right)\right] - \mathbb{E}_q\left[\ln\left(\frac{1}{\Gamma(b_l)}\xi_l^{b_l-1}e^{-\xi_l}\right)\right]$ |
| $\omega_l$ | Gamma | Generalized InvGaussian | $\mathbb{E}_q\left[\ln\left(\frac{(\rho/\chi)^{\lambda_0/2}}{2K_{\lambda_0}(\sqrt{\rho\chi})}\omega_l^{\lambda_0-1}e^{(-\rho\omega_l+\chi/\omega_l)/2}\right)\right] - \mathbb{E}_q\left[\ln\left(\frac{\xi_l^{a_l}}{\Gamma(a_l)}\omega_l^{a_l-1}e^{-\omega_l\xi_l}\right)\right]$ |
| $\psi_{jl}$ | Exp | Reciprocal InvGaussian | $\mathbb{E}_q\left[\ln\left(\frac{1}{\psi_{jl}\sqrt{2\pi}}\exp\left(\frac{(1-\psi_{jl}\mu)^2}{2\psi_{jl}\mu}\right)\right)\right] - \mathbb{E}_q\left[\ln\left(\frac{1}{2}e^{-\frac{1}{2}\psi_{jl}}\right)\right]$ |

## 5 Simulation Study

We first validate our method on simulated scenarios to validate the predictive and shrinkage performance of the R2D2-Net. We control the depth to observe how the performance varies as the depth of the network increases.

### 5.1 Experimental Setup

**Scenarios.** We generate the data $\mathcal{D} = \{\boldsymbol{x}_i, y_i\}_{i=1}^N$ with $N = 10000$ and each data point $x_{ij} \in \boldsymbol{x}_i$ is sampled from a uniform distribution $\mathcal{U}(-5, 5)$, and the noise $\epsilon_i \sim \mathcal{N}(0, 3^2)$. We design three simulation scenarios: (1) **Polynomial case**: $y_i = x_i^3 + \epsilon_i$; (2) **Low-dimensional non-linear regression**: $y_i = x_{i1}x_{i2} + x_{i3}x_{i4} + \epsilon_i$; (3) **High-dimensional non-linear regression**: $y_i = f(\boldsymbol{x}_i) + \epsilon_i$, where $f$ is a two-layer multiple layer perceptron (MLP) with randomly initialized weights and *Relu* nonlinearity. Additional scenarios and results are presented in the appendix. In contrast to the other scenarios, the data in Scenario 3 are generated from a randomly initialized neural network. The features are hence mostly noise (or trivial) features and shrinkage methods are expected to underperform as they shrink noise features to zeros.

For each scenario, we randomly generate five sets of data. We split 80% of the data as the training set and 20% as the testing set. All methods are trained with a gradient-based optimization method (i.e., *Adam*) with 100 epochs and a batch size of 1024, with possible early stopping.

### 5.2 Experimental Results

**Predictive Performance: R2D2-Net Achieves Competitive Performance with Deeper Layers.** We compare the prediction MSE and variance of each BNN design. When $L = 0$, the model is equivalent to a linear regression. Table 2 presents the simulation results, and Figure 2 shows the prediction means and variances of R2D2-Net and the baseline BNN designs. We observe that the R2D2-Net yields the smallest prediction error among all competitive designs, and the prediction variance is the

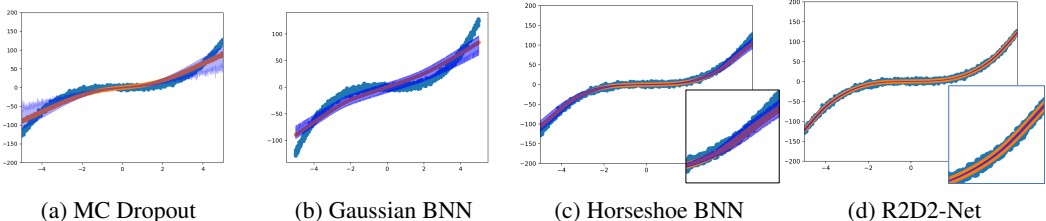

| (a) MC Dropout | (b) Gaussian BNN | (c) Horseshoe BNN | (d) R2D2-Net |

Figure 2: Test-time prediction mean and confidence interval of R2D2-Net on $y_i = x_i^3 + \epsilon_i, \epsilon_i \sim \mathcal{N}(0,9)$. The number layer is 3 and the number of samples is 100 during the validation phase. The blue dots are the ground truth data points, the yellow line is the mean of prediction and the blue shadow is the prediction interval. We observe that the R2D2-Net has a smaller prediction variance than MC Dropout, Gaussian BNN, and Horseshoe BNN.

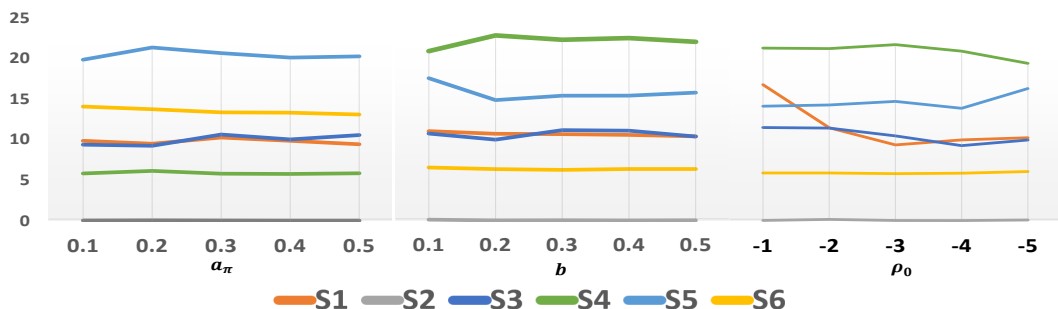

Figure 3: Ablation studies of our method to different hyperparameters. We run the six simulation scenarios (S1–S6) with an R2D2 MLP with $L = 3$, and report the testing MSEs with respect to different values of hyperparameters $a_\pi$ (left), $b$ (middle), $\rho_0$ (right).

smallest for the largest $L$. The R2D2-Net also shows greater improvement in prediction performance (i.e., smaller prediction MSE) as the number of layers increases. This demonstrates that the R2D2-Net is more capable of supporting deeper BNN architectures than other BNN designs. On the other hand, we observe that the Gaussian BNN has an increasing trend in both prediction MSE and variance as the number of layers $L$ increases. This highlights the variance inflation issues of vanilla BNNs.

**Shrinkage Performance: R2D2 Prior Best Shrinks Unnecessary Neurons to Zero.** We study the sparsification performance of R2D2-Net in comparison with the Horseshoe BNN (Ghosh et al., 2019). We investigate the distribution of the smallest node weight vectors to compare the shrinkage performance among different priors. We plot the distributions of coefficients $w_{jl}$ with the smallest magnitude $\|\mathbb{E}[w_{jl}]\|$ (Figure 4). We observe that the weights samples of the R2D2-Net have the highest concentration rate at zero compared with Horseshoe BNN and Gaussian BNN. This validates that the highest concentration rate property of the R2D2 prior also holds when generalized to neural networks. We also validate that the R2D2-Net has the best shrinkage performance than existing BNNs with other priors.

**Impact of Hyperparameters.** We investigate how sensitive the R2D2-Net is to the changes in the hyperparameters, such as $a_\pi, b$ and $\rho_0$. We perform the evaluation using the simulation scenarios in Section 5. Figure 3 presents the results using an R2D2 MLP with $L = 3$. We observe that our method is robust to changes in these hyper-parameters. The performance of R2D2-Net is more sensitive to the prior variance parameter $\rho_0$ than the hyperparameters $a_\pi, b$ of the R2D2 prior (Eq.(2)).

## 6 EXPERIMENTS ON REAL DATA

We extend the experiments to real data to further validate the capability of R2D2-Net when generalized to more realistic datasets (i.e., TinyImageNet) and larger architectures (i.e., residual nets).

### 6.1 EXPERIMENTAL SETUP

**Datasets.** We evaluate the R2D2-Net on standard computer vision datasets in comparison with existing methods. For **image classification**, we use CIFAR 10, CIFAR 100, and TinyImageNet as the benchmark datasets. We perform 5-fold cross-validation to evaluate each method. We use accuracy, macro F1 score, and area under the receiver operating curve (AUROC) as the evaluation metric, and

Table 2: Simulation results on MSE and prediction variance under the R2D2-Net compared with different BNN designs on MLP with different numbers of layers $L = 0, 1, 2, 3$. Standard deviations over five replicates are shown in brackets.

| | **Non-Trivial Features** | | | | | | | |
|---|---|---|---|---|---|---|---|---|
| | **Scenario 1: Polynomial Case** | | | | | | | |
| | $L = 0$ | | $L = 1$ | | $L = 2$ | | $L = 3$ | |
| **BNN** | **MSE** | **Variance** | **MSE** | **Variance** | **MSE** | **Variance** | **MSE** | **Variance** |
| Gauss | 2091.7 (91.1) | 0.02 (0) | 422.66 (7.8) | 0.67 (0.0) | 371.42 (11.8) | 5.38 (0.6) | 309.45 (44.5) | 18.62 (1.75) |
| MCD | 1419.1 (115.3) | 35.73 (5.6) | 185.38 (8.6) | 37.55 (2.1) | 103.28 (3.5) | 79.08 (3.8) | 80.40 (3.2) | 62.99 (3.8) |
| MFVI | 2116.4 (80.1) | 0.02 (0.0) | 427.3 (11.7) | 0.67 (0.1) | 365.8 (16.0) | 5.57 (0.2) | 271.7 (48.6) | 17.76 (1.8) |
| DE | 1729.6 (56.0) | 2.27 (1.7) | 288.7 (6.3) | 0.63 (0.3) | 12.38 (1.0) | 0.87 (0.5) | 9.11 (0.3) | 0.13 (0.1) |
| Radial | 2091.3 (91.4) | 0.02 (0) | 423.2 (8.9) | 0.36 (0.1) | 371.1 (111.4) | 4.05 (1.1) | 221.2 (111.4) | 9.35 (4.5) |
| HS-BNN | **363.21 (10.4)** | 0.06 (0.04) | 22.64 (6.84) | 0.45 (0.09) | 19.2 (12.35) | 1.79 (0.99) | 20.37 (10.33) | 2.69 (0.26) |
| **R2D2-Net** | 891.5 (148.2) | **0.0 (0.0)** | **22.04 (2.8)** | **0.02 (0)** | **9.18 (0.4)** | **0.32 (0.1)** | **10.1 (1.41)** | **0.85 (0.15)** |
| | **Scenario 2: Low-dimensional Non-linear Regression** | | | | | | | |
| | $L = 0$ | | $L = 1$ | | $L = 2$ | | $L = 3$ | |
| **Model** | **MSE** | **Variance** | **MSE** | **Variance** | **MSE** | **Variance** | **MSE** | **Variance** |
| Gauss | 1134.2 (16.93) | 0.07 (0.0) | 473.43 (15.3) | 1.08 (0.04) | 69.15 (4.5) | 8.42 (0.7) | 56.37 (6.8) | 21.76 (0.6) |
| MCD | 824.7 (49.2) | 25.57 (2.7) | 453.99 (35.3) | 29.72 (5.5) | 111.26 (5.8) | 69.95 (3.5) | 89.88 (6.6) | 59.18 (5.4) |
| MFVI | 1150.2 (60.3) | 0.08 (0.01) | 477.2 (17.0) | 1.09 (0.05) | 70.13 (10.7) | 8.7 (0.4) | 53.40 (5.8) | 18.64 (9) |
| DE | 927.7 (45.4) | 0.74 (0.3) | 440.3 (26.3) | 2.14 (0.6) | 13.32 (0.7) | 1.08 (0.1) | 10.44 (0.2) | 1.03 (0.2) |
| Radial | 1126.8 (58.9) | 0.07 (0.01) | 472.6 (17.1) | 0.49 (0.3) | 55.1 (4.4) | 2.36 (3.6) | 41.7 (3.1) | 2.57 (0.2) |
| HS-BNN | **549.92 (20.99)** | 0.05 (0.04) | 182.3 (179) | 0.48 (0.3) | 12.0 (1.0) | 0.97 (0.2) | 15.2 (2.9) | 1.97 (0.4) |
| **R2D2-Net** | 616.36 (15.63) | **0.0 (0.0)** | **86.92 (63.5)** | **0.04 (0.01)** | **9.63 (0.14)** | **0.38 (0.04)** | **9.86 (0.2)** | **1.02 (0.09)** |

| | **Trivial Features** | | | | | | | |
|---|---|---|---|---|---|---|---|---|
| | **Scenario 3: High-dimensional Non-linear Regression** | | | | | | | |
| | $L = 0$ | | $L = 1$ | | $L = 2$ | | $L = 3$ | |
| **Model** | **MSE** | **Variance** | **MSE** | **Variance** | **MSE** | **Variance** | **MSE** | **Variance** |
| Gauss | 4.79 (0.1) | 11.5 (0.2) | 4.9 (0.1) | 2.24 (0.2) | **4.77 (0.1)** | 0.76 (0.0) | 5.2 (0.3) | 0.77 (0.0) |
| MCD | 4.6 (0.1) | 0.15 (0.0) | 5.67 (0.2) | 0.51 (0.0) | 5.85 (0.1) | 0.5 (0.0) | 5.62 (0.1) | 0.47 (0.0) |
| MFVI | 4.90 (0.1) | 11.48 (0.1) | 6.98 (0.3) | 2.14 (0.2) | 5.77 (0.0) | 0.76 (0.0) | 6.00 (0.0) | 0.77 (0.0) |
| DE | 4.56 (0.1) | 0.0 (0.0) | **4.8 (0.1)** | 1.08 (0.0) | 4.89 (0.2) | 1.33 (0.0) | **4.85 (0.2)** | 1.27 (0.0) |
| Radial | 4.77 (0.1) | 11.43 (1.0) | 6.22 (0.4) | 1.47 (0.1) | 6.35 (0.2) | 0.33 (0.3) | 6.36 (0.4) | 0.17 (0.0) |
| HS-BNN | 4.64 (0.2) | 0.03 (0.0) | 6.45 (0.3) | 0.15 (0.2) | 6.39 (1.1) | 0.61 (1.3) | 6.07 (0.4) | 0.1 (0.1) |
| **R2D2-Net** | **4.55 (0.1)** | **0.0 (0.0)** | 6.3 (0.3) | **0.0 (0.0)** | 5.67 (0.1) | **0.0 (0.0)** | 5.81 (0.3) | **0.0 (0.0)** |

In Scenario 3, Gaussian BNN yields better performance as the weights are randomly initialized and contain many noises, while shrinkage BNNs underperform because they shrink the noises to zero. On the other hand, Gaussian BNNs which do not possess shrinkage parameters keep the noise features and obtain better performance.

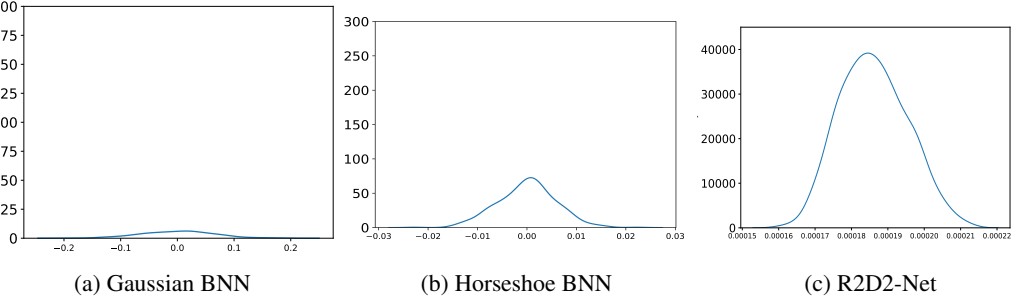

| (a) Gaussian BNN | (b) Horseshoe BNN | (c) R2D2-Net |
|---|---|---|

Figure 4: Density plots of the weight samples of Gaussian BNN, Horseshoe BNN, and R2D2-Net. We choose the weights that have the least magnitude from the first layer of a three-layer MLP. We observe that R2D2-Net has the highest concentration rate at zero.

report the mean and standard deviation of each metric. For **uncertainty estimation**, we assess the performance of the neural networks using the OOD detection task, with AUROC and the area under the precision-recall curve (AUPR) as the evaluation metrics. We treat the images in the CIFAR 10 Krizhevsky et al. (2009) dataset as the in-distribution data and the images from the fashion MNIST, OMNIGLOT, and SVHN Xiao et al. (2017) as the OOD samples. Different from some existing approaches (Malinin and Gales, 2018; Sensoy et al., 2018), we train the classifier with in-distribution only (i.e., the classifier will not see the OOD data during training).

**Competitive Methods.** We compare our method with a variety of existing BNN designs. The hyperparameter settings of each benchmark method and the summary of uncertainty measures used are presented in the supplementary materials. (1) **Frequentist CNN** (Freq): the original frequentist neural network architecture ; (2) **Gaussian BNN** (Gauss): a vanilla BNN assuming a zero-mean multivariate Gaussian as the prior distribution on the weights; (3) **Mean–field Variational Infernece**

Table 3: Image classification results of our proposed method on CIFAR 10 and CIFAR 100 with the AlexNet (Krizhevsky et al., 2012). Standard deviations are shown in brackets. **Bold** represents the best performance among BNN designs, while * represents the best performance among all models.

| Model | CIFAR 10 | | CIFAR 100 | | TinyImageNet | |
| --- | --- | --- | --- | --- | --- | --- |
| | AUROC | Accuracy | AUROC | Accuracy | AUROC | Accuracy |
| Freq | 92.70 (1.5)* | 65.03 (1.4) | 90.95 (0.2) | 31.05 (0.4) | 88.37 (1.3) | 18.30 (0.4) |
| Gauss | 91.37 (1.2) | 60.28 (1.5) | 87.24 (1.2) | 23.6 (0.6) | 87.64 (0.2) | 16.82 (0.9) |
| MCD | 90.09 (0.2) | 55.08 (0.6) | 87.67 (1.3) | 21.92 (1.1) | 86.23 (1.7) | 17.28 (1.5) |
| MFVI | 91.11 (0.9) | 59.27 (1.1) | 87.69 (0.9) | 23.06 (0.2) | 86.01 (0.3) | 12.78 (0.6) |
| Radial | 91.22 (0.8) | 63.24 (0.8) | 89.20 (1.0) | 25.70 (0.5) | 84.35 (0.6) | 12.12 (0.5) |
| DE | 90.67 (0.6) | 62.41 (0.5) | 87.97 (0.9) | 24.63 (0.4) | 86.25 (0.4) | 13.73 (0.5) |
| HS-BNN | 91.99 (0.8) | 65.01 (0.3) | 91.37 (0.2) | 33.27 (0.3) | 88.71 (2.0) | 20.33 (1.2) |
| **R2D2-Net** | **92.49 (0.2)** | **65.10 (0.02)*** | **92.48 (0.03)*** | **36.12 (0.5)*** | **88.76 (0.5)*** | **20.55 (0.4)*** |

(Blundell et al., 2015) (MFVI): classical mean-field approximation to variational distribution which assumes that it can be factorized by marginal distribution of local variables. We use the Gaussian prior as adopted in the original proposition. (4) **Horseshoe BNN** (Ghosh et al., 2019) (HS-BNN): a Bayesian variable selection method on the neural network which assumes a Horseshoe prior on the weights; (5) **MC Dropout** (Gal and Ghahramani, 2016) (MCD): using repeated dropouts on trained weights to draw Monte Carlo samples of the weights of the BNNs (reproduced from Gal and Ghahramani (2016)); (6) **RadialBNN** (Farquhar et al., 2020) (Radial): sampling from the hyperspherical coordinate system to resolve the problem in original MFVI where the probability mass is far from the true mean. (7) **Deep Ensembles** (Lakshminarayanan et al., 2017) (DE): it uses a finite ensemble of deep neural networks to approximate posterior weight distribution;

## 6.2 IMAGE CLASSIFICATION: R2D2 SHRINKAGE IMPROVES PREDICTIVE PERFORMANCE

Table 3 presents the image classification results of our R2D2-Net in comparison with existing methods. We assess our method on standard image classification benchmarks — CIFAR 10, CIFAR 100, and TinyImageNet. We fix the model architecture as AlexNet (Krizhevsky et al., 2012) for fair comparison.

Not only does our proposed method outperform the existing BNN designs, but it also occasionally outperforms the frequentist design. It is noteworthy that since BNNs impose a natural regularization on the weights, it is difficult for BNN designs to outperform their frequentist counterpart. This demonstrates that choosing the R2D2 prior can potentially lead to the best variable selection outcome. The R2D2 prior can select a suitable subset of weights with its shrinkage properties, while the frequentist design cannot. Hence, its prediction

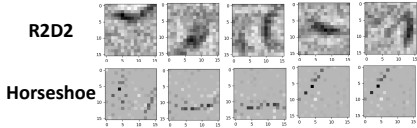

Figure 5: Five largest-norm convolutional filters of the R2D2-Net and Horseshoe BNN. We use a simple CNN with one convolutional layer and one linear layer for illustrative purposes.

performance can be more satisfactory than the original frequentist design. We visualize the five largest-norm filters of the R2D2-Net and Horseshoe BNN to compare their capabilities to select features (Figure 5). We observe that the largest filters of the R2D2-Net have more meaningful patterns, while the largest filters of the Horseshoe BNN are close to uniform noise. Since the R2D2 prior has heavier tails than the Horseshoe prior, it can preserve large signals in the filter weights and avoid over-shrinkage, as demonstrated by the difference in filter patterns in Figure 5.

## 6.3 UNCERTAINTY ESTIMATION: R2D2 SHRINKAGE CAPTURES IMPORTANT VARIANCE.

We further compare the performance of uncertainty estimation over the existing BNN designs. Additionally, we include two entropy-based uncertainty estimation methods: DPN (Malinin and Gales, 2018) and EDL (Sensoy et al., 2018), which estimate uncertainties based on the assumption of Dirichlet distribution on latent probabilities. We use the classification entropy as the uncertainty measure for each uncertainty estimation method for fair comparison. More information on baseline methods and uncertainty measures is introduced in the appendix. We adopt the OOD detection task to evaluate the performance of the R2D2-Net to estimate the uncertainty in the data. The OOD detection aims to identify whether the input data are in-distribution or from a different dataset. Table 10 presents the AUROC and the AUPR of the R2D2-Net using the classification entropy as the uncertainty measure. We treat CIFAR 10 as the in-distribution dataset and FasionMNIST, OMNIGLOT, and SVHN as the OOD datasets (more experiments with MNIST as the in-distribution dataset are presented in the appendix). We observe that our R2D2-Net demonstrates a satisfactory

Table 4: The OOD detection performance of the R2D2-Net compared with various BNN designs under the LeNet (LeCun et al., 1989), using CIFAR 10 as the in-distribution dataset. The best performance among all methods is highlighted in boldface.

| Model | Fashion MNIST | | OMNIGLOT | | SVHN | |
|---|---|---|---|---|---|---|
| | AUROC | AUPR | AUROC | AUPR | AUROC | AUPR |
| Gauss | 74.49 | 86.78 | 78.58 | 81.82 | 70.57 | 79.30 |
| HS-BNN | 80.76 | 75.99 | 86.66 | 90.18 | 70.11 | 78.39 |
| MCD | 81.80 | 74.22 | 80.03 | 82.06 | 68.58 | 78.53 |
| MFVI | 85.45 | 79.55 | 89.17 | 91.64 | 76.02 | 85.08 |
| Radial | 83.86 | 81.26 | 75.39 | 74.91 | 67.74 | 81.91 |
| DE | 71.02 | 76.81 | 86.77 | 90.35 | 61.01 | 62.59 |
| DPN | 87.07 | 83.75 | 87.07 | 83.75 | 57.48 | 77.76 |
| EDL | 89.26 | 86.16 | 66.53 | 67.12 | 69.57 | 83.74 |
| **R2D2-Net** | **92.85** | **94.09** | **91.95** | **92.25** | **79.84** | **89.24** |

performance over the baseline methods. This shows that using an R2D2 prior on the weights can effectively shrink the noises in parameters while maintaining a non-trivial variance structure. These preserved variances can be used to represent the model-wise uncertainties. Hence, the R2D2-Net can generate more accurate uncertainty estimates than existing Bayesian and non-Bayesian approaches.

## 6.4 ABLATION ANALYSIS: SATISFACTORY PERFORMANCE WITH VARIOUS ARCHITECTURES

We further apply R2D2 layers to different neural network architectures to evaluate the performance. We choose LeNet (LeCun et al., 1989) and AlexNet (Krizhevsky et al., 2012) to benchmark the performance of different BNN designs. Table 5 presents the results on CIFAR 10. We can observe that for most architectures our proposed BNN design obtains SOTA performance compared with existing BNN methods. This shows the R2D2-Net can perform satisfactorily on different architectures including modern architectures at scale (e.g., ResNet101).

Table 5: Image classification results of our proposed BNN design on different model architectures compared to different SOTA BNN designs with the CIFAR 10 dataset.

| Model | LeNet | | AlexNet | | ResNet50 | | ResNet101 | |
|---|---|---|---|---|---|---|---|---|
| | AUROC | ACC | AUROC | ACC | AUROC | ACC | AUROC | ACC |
| Freq | 91.24 | 61.21 | 92.70* | 65.03 | 96.23 | 79.25* | 96.75 | 79.20 |
| Gauss | 91.31 | 60.03 | 91.21 | 62.64 | 95.59 | 73.62 | 95.53 | 73.34 |
| MCD | 91.50 | 58.76 | 91.21 | 62.76 | 96.44 | 77.24 | 96.83 | **79.54*** |
| MFVI | 92.41 | 63.39 | 91.11 | 59.27 | 96.48 | 78.19 | 95.65 | 73.37 |
| Radial | 91.74 | 61.29 | 91.22 | 63.24 | 95.39 | 74.03 | 96.34 | 72.99 |
| DE | 93.75 | 63.11 | 90.06 | 62.94 | 96.60 | 77.62 | 96.82 | 79.01 |
| HS-BNN | **92.42*** | 60.13 | 91.99 | 65.01 | 96.96 | 78.90 | 97.08 | 79.14 |
| **R2D2-Net** | 90.43 | **61.53*** | **92.49** | **65.10*** | **96.97*** | **79.10** | **97.12*** | 79.20 |

## 7 CONCLUSION

In this work, we propose a novel BNN design — the R2D2-Net. We develop a variational Gibbs inference algorithm to better approximate the posterior distributions of weights. Extensive experiments on synthetic and real datasets validate the performance of our proposed BNN design on both image classification and image uncertainty estimation tasks.

**Limitations and Future Work.** Our proposed method tackles the scalability constraints of Bayesian neural networks and is validated on some modern architectures (e.g., residual-based). Due to the lack of mature research in Bayesian designs of more modern architectures (such as Bayesian attention mechanisms and transformers), the extension of R2D2-Net to these architectures would be non-trivial although it potentially opens the possibility of Bayesian foundation models. On the other hand, we take Monte Carlo samples of weights from the posterior distribution, which could potentially be a computational burden. Integration of recent efficient sampling techniques of BNN (Dusenberry et al., 2020) would decrease the posterior inference complexity. Our proposed method can be potentially applied to data domains, such as graphs. The R2D2-Net also has great real application potential in reinforcement learning, recommendation systems, and biomedical imaging for its capability in predictive inference and uncertainty estimation.

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

**Overview.** In the appendix, we first highlight the differnece between our work and Zhang et al. in Section A. We present a detailed summary of the datasets in Section B. We demonstrate additional experiment results of the R2D2-Net, and more details on implementations and hyperparameters in Section C. We also provide further details on composing NN architectures with R2D2 layers and illustrative architectural samples of the R2D2-Net in Section C.3. Additionally, we provide the properties of distributions used (Section D), and information on other global–local shrinkage priors in Section F. We also provide detailed derivations of the KL divergences introduced in the main text in Section E. Furthermore, we provide a detailed algorithm of R2D2-Net in Algorithm 1. Finally, we provide detailed definitions of the evaluation metrics (Section H) and uncertainty measures (Section G) used in experiments.

## A    DIFFERENCE BETWEEN OUR WORK AND ZHANG ET AL.

We additionally highlight the difference between our work and Zhang et al.. Essentially our work borrowed the prior developed by Zhang et al. and adapted the Gibbs inference algorithm from the paper to the deep learning context. We consider R2D2 as an important prior with several decent properties (the highest concentration rate at zero and the heaviest tails) that are crucial to the development of BNN models. A well-known work is the Horseshoe BNN (Ghosh et al., 2019) extending the Horseshoe prior (Carvalho et al., 2009), which enables Horseshoe to be a major prior in sparsifying neural networks. We aim to highlight the disadvantages (i.e., the heavier tail and lower concentration rate at zero) of the Horseshoe prior and show that the R2D2 prior is a better choice for variable shrinkage in neural networks.

# B  ADDITIONAL INFORMATION ON DATASETS

We present more details of the datasets used for experiments. Table 6 provides a summary of the datasets. We resize the images to $32 \times 32$ for classification and out-of-distribution (OOD) detection tasks.

Table 6: Summary of Datasets

| Dataset | No. Classes | No. Training | No. Testing |
|---|---|---|---|
| **MNIST** | 10 | 60,000 | 10,000 |
| **Fashion-MNIST** | 10 | 60,000 | 10,000 |
| **OMNIGLOT** | 50 | 13,180 | 19,280 |
| **SVHN** | 10 | 73,257 | 26,032 |
| **CIFAR-10** | 10 | 60,000 | 10,000 |
| **CIFAR-100** | 100 | 60,000 | 10,000 |
| **TinyImageNet** | 200 | 80,000 | 20,000 |

# C  ADDITIONAL INFORMATION ON EXPERIMENTS

We demonstrate additional information on baselines and experiment results in this section, including additional settings of hyperparameters and implementations.

## C.1  ADDITIONAL INFORMATION ON BASELINES

In addition to existing BNN designs, we add two entropy-based uncertainty estimation methods for comparison. Since we are using entropy as the uncertainty metric (as this is a classic metric for classification uncertainty) for a fare comparison, the OOD performance may be slightly worse than their respective state-of-the-art performance.

(1) **DPN** (Malinin and Gales, 2018): it assumes a Dirichlet distribution on the classification output and trains an OOD classifier by minimizing the KL divergences between the prior and posterior distributions; (2) **EDL** (Sensoy et al., 2018): in addition to DPN (Malinin and Gales, 2018), EDL trains the classifier with the cross-entropy loss and the KL divergence between the prior and posterior distributions.

## C.2  ADDITIONAL EXPERIMENT RESULTS

**Additional Image Classification Results.** We present additional results on image classification and ablation studies with different architectures using more evaluation metrics. Table 7 presents the image classification results on CIFAR 10 with the architecture fixed as LeNet. We observe that the improvement of R2D2-Net is less significant compared to that when AlexNet is used.

Table 7: Image classification results of our proposed method on CIFAR 10 and CIFAR 100 with the LeNet (LeCun et al., 1989). Standard deviations are shown in brackets. We report the Macro-F1 in addition to the AUROC and Accuracy reported in the main text.

| | CIFAR 10 | | | CIFAR 100 | | |
|---|---|---|---|---|---|---|
| **Model** | **AUROC** | **Accuracy** | **Macro-F1** | **AUROC** | **Accuracy** | **Macro-F1** |
| Frequentist | 91.38 | 62.24 | 62.34 | 89.45 | 30.51 | 29.64 |
| Gaussian BNN | 91.31 | 60.03 | 59.55 | 89.17 | 25.79 | 25.08 |
| MC Dropout | 91.50 | 58.76 | 59.4 | 90.65 | 27.23 | 25.83 |
| MFVI | 92.22 | 61.94 | 61.71 | 88.90 | 29.63 | 29.06 |
| Radial BNN | 92.13 | 61.71 | 61.30 | 89.77 | 30.27 | 29.8 |
| Deep Ensembles | 92.74 | 64.26 | 64.14 | 89.37 | 30.04 | 29.44 |
| Horseshoe BNN | 92.42 | 60.13 | 59.80 | 85.88 | 17.94 | 16.01 |
| **R2D2-Net (ours)** | 92.39 | 62.14 | 62.02 | 88.59 | 30.51 | 29.82 |

Table 8: Image classification results of our proposed method on CIFAR 10 and CIFAR 100 with the AlexNet (Krizhevsky et al., 2012). Standard deviations are shown in brackets. We report the Macro-F1 in addition to the AUROC and Accuracy reported in the main text. **Bold** represents the best performance among BNN designs, while * represents the best performance among all models.

| Model | CIFAR 10 | | | CIFAR 100 | | |
|---|---|---|---|---|---|---|
| | AUROC | Accuracy | Macro-F1 | AUROC | Accuracy | Macro-F1 |
| Frequentist NN | 92.70 (1.5)* | 65.03 (1.4) | 64.9 (1.6)* | 90.95 (0.2) | 31.05 (0.4) | 31.21 (0.6) |
| Gaussian BNN | 91.37 (1.2) | 60.28 (1.5) | 60.78 (1.4) | 87.24 (1.2) | 23.60 (0.6) | 22.02 (0.8) |
| MC Dropout | 90.09 (0.2) | 55.08 (0.6) | 54.22 (0.3) | 87.67 (1.3) | 21.92 (1.1) | 19.74 (1.1) |
| MFVI | 91.11 (0.9) | 59.27 (1.1) | 61.82 (0.5) | 87.69 (0.9) | 23.06 (0.2) | 22.34 (0.2) |
| Radial BNN | 91.22 (0.8) | 63.24 (0.8) | 62.48 (0.9) | 89.20 (1.0) | 25.70 (0.5) | 24.89 (0.7) |
| Deep Ensembles | 90.67 (0.6) | 62.41 (0.5) | 62.43 (0.4) | 87.97 (0.9) | 24.63 (0.4) | 23.94 (0.5) |
| Horseshoe BNN | 91.99 (0.8) | 65.01 (0.3) | 64.7 (0.3) | 91.37 (0.2) | 33.27 (0.3) | 34.02 (0.3) |
| **R2D2-Net** | **92.49 (0.2)** | **65.10 (0.02)** | **65.14 (0.06)** | **91.41 (0.03)*** | **36.12 (0.5)*** | **34.83 (0.4)*** |

**Additional Simulation Scenarios and Results.** We more simulation scenarios and results in this section. We first consider the sparsity in the coefficients: (4) **Sparse coefficients**: $y_i = \boldsymbol{x}_i^\top \boldsymbol{\beta} + \epsilon_i$, where 90% of the coefficients in $\boldsymbol{\beta}$ are set to be 0; (5) **Dense coefficients**: $y_i = \boldsymbol{x}_i^\top \boldsymbol{\beta} + \epsilon_i$, where 10% of the coefficients in $\boldsymbol{\beta}$ are set to be 0. We then consider a multiple linear regression case where the response is a linear combination of covariates: (6) **Linear regression**: $y_i = \boldsymbol{x}_i^\top \boldsymbol{\beta} + \epsilon_i$ where $\boldsymbol{\beta}, \boldsymbol{x}_i \in \mathbb{R}^p$. Table 9 presents the results.

Table 9: Simulation results on MSE and prediction variance under the R2D2-Net compared with different BNN designs on MLP with different numbers of layers $L = 0, 1, 2, 3$. Standard deviations over five replicates are shown in brackets.

| | Scenario 4: Multiple Linear Regression | | | | | | | |
|---|---|---|---|---|---|---|---|---|
| | $L = 0$ | | $L = 1$ | | $L = 2$ | | $L = 3$ | |
| Model | MSE | Variance | MSE | Variance | MSE | Variance | MSE | Variance |
| Gaussian | 478.28 (8.41) | 0.08 (0.01) | 80.7 (5.98) | 0.57 (0.03) | 3.87 (8.41) | 3.82 (0.18) | 4.11 (1.84) | 5.27 (0.41) |
| MC Dropout | 397.81 (20.81) | 6.85 (0.57) | 15.29 (0.24) | 9.12 (0.74) | 16.9 (0.84) | 14.25 (0.77) | 13.47 (0.69) | 11.68 (0.64) |
| Horseshoe | 392.82 (10.64) | 0 (0) | 3.84 (1.02) | 0.09 (0.03) | 1.57 (0.92) | 0.25 (0.08) | 0.57 (0.42) | 0.44 (0.05) |
| **R2D2-Net** | **316.19 (8.43)** | **0 (0)** | **1.49 (0.26)** | **0.02 (0)** | **0.03 (0.02)** | **0.07 (0)** | **0.02 (0.01)** | **0.12 (0.01)** |
| | Scenario 5: Sparse Coefficients | | | | | | | |
| | $L = 0$ | | $L = 1$ | | $L = 2$ | | $L = 3$ | |
| Model | MSE | Variance | MSE | Variance | MSE | Variance | MSE | Variance |
| Gaussian | 40.06 (3.24) | 14.15 (0.09) | 19.6 (0.82) | 12.17 (0.42) | 17.48 (1.09) | 11.62 (0.22) | 19.55 (2.5) | 11.82 (0.93) |
| MC Dropout | 66.91 (2.09) | 50 (1.55) | 13.7 (0.32) | 6.26 (0.2) | 14.36 (0.27) | 5.74 (0.19) | 14.16 (0.22) | 4.97 (0.22) |
| Horseshoe | 4.88 (0.16) | 0.64 (0.69) | 17.22 (10.26) | 1.44 (2.53) | 15.66 (1.18) | 0.52 (0.26) | 14.12 (1.32) | 0.63 (0.45) |
| **R2D2-Net** | **4.58 (0.2)** | **0.31 (0)** | **14.66 (0.5)** | **0.36 (0.01)** | **13.93 (0.41)** | **0.35 (0.01)** | **13.58 (0.48)** | **0.39 (0.01)** |
| | Scenario 6: Dense Coefficients | | | | | | | |
| | $L = 0$ | | $L = 1$ | | $L = 2$ | | $L = 3$ | |
| Model | MSE | Variance | MSE | Variance | MSE | Variance | MSE | Variance |
| Gaussian | 396.94 (9.88) | 17.65 (0.11) | 25.83 (0.34) | 31.18 (0.43) | 42.57 (3.87) | 48.18 (1.73) | 47.24 (2.67) | 62.94 (3.16) |
| MC Dropout | 516.61 (12.78) | 409.22 (10.52) | 53.22 (1.27) | 41.66 (1.36) | 66.34 (1.87) | 49.93 (2.44) | 65.68 (6.28) | 41.05 (4.21) |
| Horseshoe | 6.2 (3.34) | 3.4 (5.15) | 20.44 (10.83) | 1.39 (0.57) | 34.97 (12.25) | 3.36 (1.47) | 38.92 (10.34) | 4.05 (2.5) |
| **R2D2-Net** | **4.7 (0.11)** | **0.36 (0)** | **14.73 (0.23)** | **0.88 (0.07)** | **17.64 (1.19)** | **1.46 (0.04)** | **19.4 (1.08)** | **2.05 (0.07)** |

**Additional Results on OOD Detection.** We include additional results on OOD detection with MNIST as the in-distribution dataset and FashionMNIST, OMNIGLOT, and SVHN as the OOD dataset (see Table 10). We observe that the R2D2-Net still obtains an ideal uncertainty estimation performance under this setting.

**Implementation Details and Hyperparameters** The proposed method is implemented in Python with *Pytorch* library on a server equipped with four NVIDIA TESLA V100 GPUs. All methods are trained for 1000 epochs for image classification and 100 epochs for OOD detection with possible early stopping. We randomly initialize the weights of each architecture (i.e., train from scratch). We select the checkpoint which has the largest validation AUROC as the testing checkpoint. We use *Adam* as the optimizer with a learning rate of 0.0005 and weight decay of 0.0005. The batch size is 1024. The dropout ratio is 0.2 for MC Dropout (Gal and Ghahramani, 2016). We set a universal annealing rate of 0.001 for the KL loss since we did not encounter KL vanishing problem. Data

Table 10: The OOD detection performance of the R2D2-Net compared with various BNN designs under the LeNet (LeCun et al., 1989), using MNIST as the in-distribution dataset. The best performance among all methods is highlighted in boldface.

| Model | Fashion MNIST | | OMNIGLOT | | SVHN | |
| --- | --- | --- | --- | --- | --- | --- |
| | AUROC | AUPR | AUROC | AUPR | AUROC | AUPR |
| Gauss | 98.36 | 98.36 | 99.17 | 99.38 | 98.95 | 99.10 |
| HS-BNN | 80.76 | 75.99 | 99.06 | 99.65 | 99.35 | 98.74 |
| MCD | 81.80 | 74.22 | 80.03 | 82.06 | **99.96** | **99.96** |
| MFVI | 98.52 | 98.48 | 98.94 | 99.11 | 99.91 | **99.96** |
| Radial | 98.2 | 97.94 | 98.52 | 98.73 | 99.64 | 99.85 |
| DE | 90.70 | 91.08 | 99.70 | 91.08 | 99.21 | 99.68 |
| DPN | 98.70 | 98.80 | 99.96 | 99.96 | **99.96** | **99.96** |
| EDL | 73.43 | 80.22 | 72.61 | 81.42 | 63.43 | 85.09 |
| **R2D2-Net** | **98.75** | **98.84** | **99.64** | **99.65** | 99.31 | 99.69 |

augmentations such as colour jittering and random cropping and flipping are applied to regularize the learning process.

**Learning Curve Comparison**

We compare the learning curves from different BNN designs in Figure 6. We observe that the R2D2 Net has consistent higher testing performance from epoch to epoch.

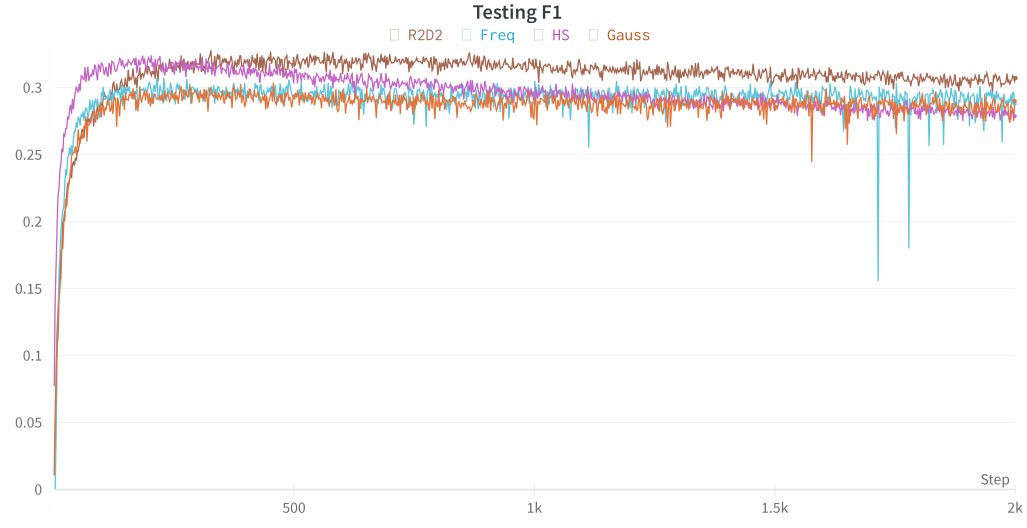

Figure 6: Training Curve Comparison.

## C.3 ADDITIONAL DETAILS ON MODEL ARCHITECTURES

**Compose Network Architecture with R2D2 Layers.** With the marginal distributions of weights in Eq. (2), we can formulate the layers of the R2D2-Net. Specifically, we consider two basic operations in a neural network — the linear layer and the convolutional layer. Let $w_l$ be the vector of all weight parameters of the $l$-th layer. The distribution of the $j$-th element $w_{jl}$ follows the R2D2 distribution given in Eq. (2). We compose the neural network architecture by specifying a combination of convolutional layers and linear layers. Figure 1 presents the conditional dependencies of the R2D2 design and the training paradigm. As an illustrative example, the visualization of R2D2 LeNet is provided in the appendix. Each linear layer and convolutional layer are replaced by the R2D2 counterparts (i.e., the R2D2 Linear and R2D2 Conv), while the pooling layers and activation layers remain the same as their frequentist designs.

**Summary of Model Architectures and Complexity.** We summarize the model architectures used in experiments and their complexity in Table 11. Figure 7 presents an example of the LeNet (LeCun et al., 1989) architecture composed by R2D2 layers, where each convolutional layer and each linear layer are replaced by its corresponding BNN design (e.g., R2D2 linear layer or R2D2 Conv Layer).

Table 11: Summary of models used in experiments and their depth. F stands for the frequentist network, and B stands for the Bayesian counterpart.

| Model | # Params (F) | # Params (B) |
|---|---|---|
| LeNet | 62K | 124K |
| AlexNet | 2.8M | 5.6M |
| ResNet50 | 25.6M | 51.2M |
| ResNet101 | 44.5M | 89M |

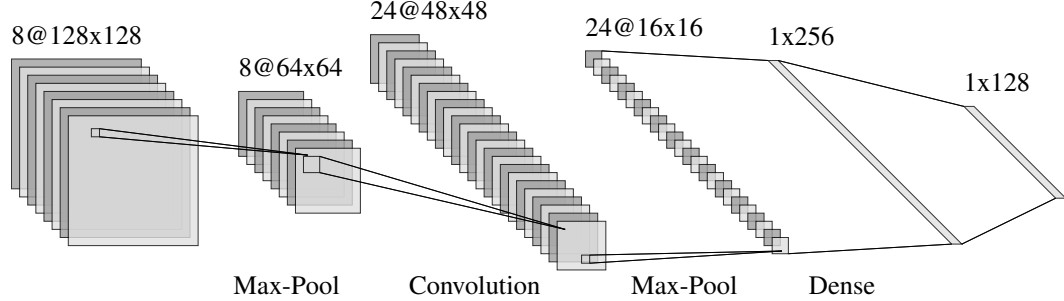

Figure 7: Example of the R2D2 LeNet architecture. Each convolutional or linear layer is replaced by its R2D2 design (i.e., R2D2 Linear or R2D2 Conv).

### C.4 HYPERPARAMETER SETTINGS

The hyperparameter settings for different priors are as follows

- Number of posterior samples (during inference): 100
- Gaussian BNN:
    - $\rho_0 \sim \mathcal{N}(-3, 0.1^2)$
- Horseshoe
    - Global shrinkage $b_g = 1.0$
    - Local Shrinkage $b_0 = 1.0$
    - $\rho_0 \sim \mathcal{N}(-3, 0.1^2)$
- R2D2
    - $a_\pi = 0.6$
    - $b = 0.5$
    - prior mean of $w_{jl} = 0$
    - $\rho_0 \sim \mathcal{N}(-3, 0.1^2)$

## D DISTRIBUTIONS

The Multivariate Gaussian and Generalized Inverse Gaussian distributions are crucial for posterior computations, we provided the formal definitions and their important properties in this section.

**Multivariate Gaussian.** The density of a multivariate Gaussian distribution is defined as

---

**Algorithm 1** The Variational Gibbs Inference Algorithm.

---

**Input:**
Number of layers $L$; Prior distributions of weight parameters $\pi(\boldsymbol{\theta})$;
Number of steps $n_s$; Total numbers of parameters of each layer $\{p_l\}_{l=1}^L$;
Local shrinkage parameters $\boldsymbol{\phi} = \{\phi_{jl}\}_{j=1}^{p_l}{}_{l=1}^L, \boldsymbol{\psi} = \{\psi_{jl}\}_{j=1}^{p_l}{}_{l=1}^L$;
Global shrinkage parameters $\boldsymbol{\omega} = \{\omega_l\}_{l=1}^L, \boldsymbol{\xi} = \{\xi_l\}_{l=1}^L$;
Input data $\mathcal{D} = \{\boldsymbol{x}_i, y_i\}_{i=1}^N$;
Hyperparameters $a_\pi, \rho_0, b$.
**Output:**
The posterior distribution of the weights $p(\boldsymbol{\theta}|\mathcal{D})$.
1: Initialize $\pi(w_{jl}) \sim \mathcal{N}(\mathbf{0}, (\log(1 + e^{\rho_0}))^2), a_l = p_l a_\pi, b_l = b, \mu_{w_{jl}} = 0$
2: Sample $\psi_{jl} \sim \text{Exp}(1/2), \phi_{jl} \sim \text{Dir}(a_\pi, \ldots, a_\pi), \xi_l \sim \text{Ga}(b_l, 1), \omega_l|\xi_l \sim \text{Ga}(a_l, \xi_l)$
3: **for** $s$ in $1 : n_s$ **do**
4:    **for** $l$ in $1 : L$ **do**
5:        **for** $w_{jl}$ in $\boldsymbol{w}_l$ **do**
6:            Sample $w_{jl} \sim \mathcal{N}(\mu_{w_{jl}}, \psi_{jl}\phi_{jl}\omega_l\sigma_{jl}^2/2)$ $\qquad\qquad\qquad$ ▷ Sample weights
7:            Sample $\rho_{jl} \sim \mathcal{N}(\mu_{\rho_{jl}}, \sigma_{\rho_{jl}}^2)$
8:            Set $\sigma_{jl} = \log(1 + e^{\rho_{jl}})$ $\qquad\qquad\qquad\qquad$ ▷ Reparameterized Gaussian
9:            Sample $\omega \sim \text{giG}(\chi = \sum_{j=1}^{p_l} 2w_{jl}^2/(\sigma^2\phi_{jl}\psi_{jl}), \rho = 2\xi_l, \lambda_0 = a_l - \frac{p_l}{2})$
10:           Sample $\xi_l \sim \text{Ga}(a_l + b_l, 1 + \omega_l)$
11:           Sample $\psi_{jl}^{-1} \sim \text{InvGaussian}(\mu = \sqrt{\sigma_{jl}^2\phi_{jl}\omega_l/2}/|w_{jl}|, \lambda = 1)$
12:           Sample $T_{jl} \sim \text{giG}(\chi = 2w_{jl}^2/(\sigma_{jl}^2\psi_{jl}), \rho = 2\xi_l, \lambda_0 = a_l - \frac{p_l}{2})$
13:           Set $\phi_{jl} = T_{jl}/\sum_j T_{jl}$
14:       **end for**
15:       **if** $l == 1$ **then**
16:           Compute $\boldsymbol{h}_{l+1} = \boldsymbol{w}_l\boldsymbol{x}_n + \text{bias}_l$ $\qquad\qquad\qquad\qquad$ ▷ Hidden features
17:       **else**
18:           Compute $\boldsymbol{h}_{l+1} = \boldsymbol{w}_l\boldsymbol{h}_l + \text{bias}_l$
19:       **end if**
20:    **end for**
21:    Obtain prediction $\hat{y}_n$ from $h_{L-1}$
22:    Compute the supervision loss, $\text{KL}(q\|\pi)$ (Table 1 in the main text), and the ELBO.
23:    Backpropagate the ELBO to update the mean and variance of $\boldsymbol{\theta}$.
24: **end for**
25: **return** The posterior distribution $p(\boldsymbol{\theta}|\mathcal{D})$.

---

$$p(\boldsymbol{x}; \boldsymbol{\mu}, \boldsymbol{\Sigma}) = \frac{1}{(2\pi)^{\frac{p}{2}}|\boldsymbol{\Sigma}|^{\frac{1}{2}}} \exp\left\{-\frac{1}{2}(\boldsymbol{x} - \boldsymbol{\mu})^\top \boldsymbol{\Sigma}^{-1}(\boldsymbol{x} - \boldsymbol{\mu})\right\},$$

where $\boldsymbol{\mu} \in \mathbb{R}^p$ is the mean vector and $\boldsymbol{\Sigma} \in \mathbb{R}^{p\times p}$ is the covariance matrix.

**Generalized Inverse Gaussian.** Denote $Z \sim \text{giG}(\chi, \rho, \lambda_0)$, the generalized inverse Gaussian distribution, which has the density function $f(z) = \frac{(\rho/\chi)^{\frac{\lambda_0}{2}}}{2K_{\lambda_0}(\sqrt{\rho\chi})} z^{\lambda_0-1} \exp\{-(\rho z + \chi/z)/2\}$, where $K_{\lambda_0}$ is a modified Bessel function of the second kind. Specifically, an inverse Gaussian distribution of the form $f(x; \mu, \lambda) = \left(\frac{\lambda}{2\pi x^3}\right)^{1/2} \exp\left(\frac{-\lambda(x-\mu)^2}{2\mu^2 x}\right)$ is a giG with $\rho = \lambda/\mu^2, \chi = \lambda$, and $\lambda_0 = -\frac{1}{2}$.

**Expectations of Generalized Inverse Gaussian.** We provide the well-known results of the expectations of functions of $X \sim \text{giG}(\chi, \rho, \lambda_0)$ here for completeness:

$$\mathbb{E}(X) = \frac{\sqrt{\chi}K_{\lambda_0+1}(\sqrt{\rho\chi})}{\sqrt{\lambda_0}K_{\lambda_0+1}(\sqrt{\rho\chi})}$$

$$\mathbb{E}\left(\frac{1}{X}\right) = \frac{\sqrt{\rho}K_{\lambda_0+1}(\sqrt{\rho\chi})}{\sqrt{\chi}K_{\lambda_0+1}(\sqrt{\rho\chi})} - \frac{2\lambda_0}{\chi}$$

$$\mathbb{E}(\ln X) = \ln\frac{\sqrt{\chi}}{\sqrt{\rho}} + \frac{\partial}{\partial\lambda_0}\ln K_{\lambda_0}(\sqrt{\rho\chi}).$$

The derivative term in the above equation does not have an analytical form and therefore needs to be computed numerically.

## E  KULLBACK–LEIBLER (KL) DIVERGENCE

We provide detailed derivations of the KL divergences introduced in the main text.

**KL Divergence of Gamma Distributions.** Define the integral

$$I(a,b,c,d) = \int_0^\infty \log\left(\frac{e^{x/a}x^{b-1}}{a^b\Gamma(b)}\right)\frac{e^{x/c}x^{d-1}}{c^d\Gamma(d)}dx,$$

and then we have

$$I(a,b,c,d) = -\frac{cd}{a} - \log(a^b\Gamma(b)) + (b-1)\psi(d) + (b-1)\log(c), \tag{3}$$

where $\psi$ is the digamma function. The KL divergence between two Gamma distributions can be obtained in a closed form as

$$\mathrm{KL}(\mathrm{Ga}(a,b)\|\mathrm{Ga}(c,d)) = I(a,b,c,d) - I(c,d,c,d)$$

**KL Divergence of Multivariate Normal Distributions.** The KL divergence of two multivariate normal distributions $\mathcal{N}(\boldsymbol{\mu}_1,\boldsymbol{\Sigma}_1)$ and $\mathcal{N}(\boldsymbol{\mu}_2,\boldsymbol{\Sigma}_2)$ is

$$\mathrm{KL}(\mathcal{N}(\boldsymbol{\mu}_1,\boldsymbol{\Sigma}_1)\|\mathcal{N}(\boldsymbol{\mu}_2,\boldsymbol{\Sigma}_2)) = \frac{1}{2}\left[\log\frac{|\boldsymbol{\Sigma}_2|}{|\boldsymbol{\Sigma}_1|} - p + \mathrm{tr}\{\boldsymbol{\Sigma}_2^{-1}\boldsymbol{\Sigma}_1\} + (\boldsymbol{\mu}_2-\boldsymbol{\mu}_1)^\top\boldsymbol{\Sigma}_2^{-1}(\boldsymbol{\mu}_2-\boldsymbol{\mu}_1)\right]$$

**KL Divergence of Shrinkage Parameters.** The closed form of $\mathrm{KL}(q(\xi|\cdot)\|\pi(\xi))$ is given by

$$\begin{aligned}
\mathrm{KL}(q(\xi|\cdot)\|\pi(\xi)) &= \mathbb{E}_{q(\xi|\cdot)}[\ln q(\xi|\cdot) - \ln\pi(\xi)]\\
&= \mathbb{E}_q\left[\ln\left(\frac{(1+\omega_l)^{a_l+b_l}}{\Gamma(a_l+b_l)}\xi_l^{a_l+b_l-1}e^{-(1+\omega_l)\xi_l}\right)\right] - \mathbb{E}_q\left[\ln\left(\frac{1}{\Gamma(b_l)}\xi_l^{b_l-1}e^{-\xi_l}\right)\right]\\
&= I(a_l+b_l, 1+\omega_l, 1, b_l) - I(1, b_l, 1, b_l),
\end{aligned}$$

where the integral $I$ is defined by Eq. (3).

The closed form of $KL(q(\omega_l|\cdot)\|\pi(\omega_l))$ is given by:

$$\begin{aligned}
\mathrm{KL}(q(\omega_l|\cdot)\|\pi(\omega_l)) &= \mathbb{E}_{q(\omega_l|\cdot)}[\ln q(\omega_l|\cdot) - \ln\pi(\omega|\xi_l)]\\
&= \mathbb{E}_q\left[\ln\left(\frac{(\rho/\chi)^{\lambda_0/2}}{2K_{\lambda_0}(\sqrt{\rho\chi})}\omega_l^{\lambda_0-1}e^{(-\rho\omega_l+\chi/\omega_l)/2}\right)\right] - \mathbb{E}_q\left[\ln\left(\frac{\xi_l^{a_l}}{\Gamma(a_l)}\omega_l^{a_l-1}e^{-\omega_l\xi_l}\right)\right]\\
&= \frac{\lambda_0}{2}\ln\frac{\rho}{\chi} - \ln 2 - \ln K_{\lambda_0}(\sqrt{\rho\chi}) + (\lambda_0-1)\mathbb{E}[\ln\omega_l] - \frac{1}{2}\mathbb{E}(\rho\omega_l + \frac{\chi}{\omega_l})\\
&\quad - \rho\ln\xi + \ln\Gamma(a_l) - (a_l-1)\mathbb{E}[\ln\omega_l] + \xi\omega_l
\end{aligned}$$

The closed form of the KL divergence of $\psi_{jl}$ is given by

$$KL(q(\psi_{jl}|\cdot)\|\pi(\psi_{jl})) = \mathbb{E}_{q(\psi_{jl}|\cdot)}[\ln q(\psi_{jl}|\cdot) - \ln \pi(\psi_{jl})]$$

$$= \mathbb{E}_{q(\psi_{jl}|\cdot)}\left[\ln\left(\frac{1}{\psi_{jl}\sqrt{2\pi}}\exp\left(\frac{(1-\mu\psi_{jl})^2}{2\psi_{jl}\mu}\right)\right)\right] - \mathbb{E}_{q(\psi_{jl}|\cdot)}\left[\ln\left(\frac{1}{2}e^{-\frac{1}{2}\psi_{jl}}\right)\right]$$

$$= \mathbb{E}_{q(Y|\cdot)}\left[\ln Y + \ln(\frac{1}{2\pi}) + \frac{Y(1-\frac{\mu}{Y})^2}{2\mu} - \ln\frac{1}{2} + \frac{1}{2Y}\right]$$

where the third equation holds by introducing $Y = \dfrac{1}{\psi_{jl}} \sim \text{InvGaussian}$. The above expression can be solved by using the expectations of inverse Gaussian.

## F  GLOBAL–LOCAL SHRINKAGE PRIORS

Figure 8 presents the comparison of marginal densities of typical global–local shrinkage priors. Table 12 presents the comparisons of the concentration rate at 0 and tail thickness of typical global–local shrinkage priors. Table 12 and Figure 8 demonstrate that the R2D2 prior has the highest concentration rate at zero and the heaviest tail. The rates in Table 12 can be derived from the density functions of the global–local shrinkage priors, we are provided in the following subsections.

**The Horseshoe Prior.**
$$\beta_j|\tau_j \sim \mathcal{N}(0, \tau_j^2) \text{ where } \tau \sim C^+(0, b_0)$$

where $C^+$ is the Half-Cauchy distribution.

**The Horseshoe+ Prior.**
$$\beta_j|\tau_j \sim \mathcal{N}(0, \tau_j^2) \text{ where } \tau_j|\lambda, \eta_j \sim C^+(0, \lambda\eta_j), \ \eta_j \sim C^+(0, 1).$$

**The Dirichlet Laplace Prior.** The Dirichlet–Laplace prior (Bhattacharya et al., 2015) is given by
$$\beta_j|\phi_j \sim \text{DE}(\phi_j), \phi_j \sim \text{Ga}(a^*, 1/2).$$

**The Generalized Double Pareto Prior.** The density of the generalized double Pareto prior is
$$\pi_{\text{GDP}}(\beta_j|\eta, \alpha) = (1 + |\beta_j|/\eta)^{-\alpha+1}/(2\eta/\alpha), \quad (\alpha, \eta > 0).$$

**Alternative Form of the R2D2 Prior.** The alternative form of the R2D2 prior allows for an alternative formulation of the variational Gibbs inference paradigm, which is provided below

$$\beta_j \mid \sigma^2, \phi_j, \omega \sim \text{DE}(\sigma(\phi_j\omega/2)^{\frac{1}{2}}), \ \phi \sim \text{Dir}(a_\pi, \ldots, a_\pi), \ \omega \sim \text{BP}(a, b),$$

where BP denotes the beta-prime distribution, DE denotes the double-exponential distribution, and Dir denotes the Dirichlet distribution.

## G  UNCERTAINTY MEASURES

We describe the uncertainty measures used in the OOD misclassification task in this section. The definitions are well-known and summarized by Malinin and Gales (Malinin and Gales, 2018),

- Entropy:
$$H[p(\boldsymbol{\mu}|\mathcal{D})] = -\int_{S^{K-1}} p(\boldsymbol{\mu}|\mathcal{D})\ln p(\boldsymbol{\mu}|\mathcal{D})d\boldsymbol{\mu},$$
where $v_j$ is the normalized prediction score for class $j$.
- Maximum probability: we take the maximum predicted probability $\mathcal{P}$ from all classes as the confidence score,
$$\mathcal{P} = \max_c P(w_c|\mathcal{D}).$$
where $P(w_c|\mathcal{D})$ is the predicted probability for class $c$.

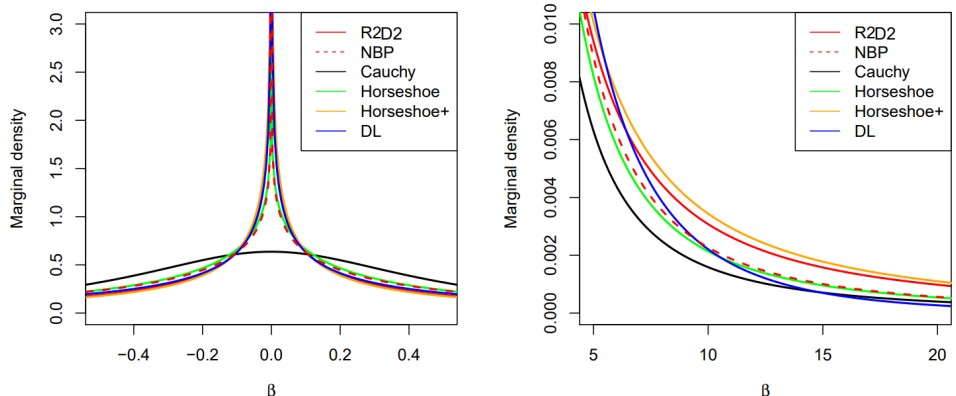

Figure 8: Marginal densities of typical global–local shrinkage priors (Zhang et al., 2020)

Table 12: Tail decay and concentration at zero of Global–local shrinkage priors (Zhang et al., 2020)

| Priors | Tail Decay | Concentration at 0 |
|---|---|---|
| Horseshoe | $\mathcal{O}\left(\dfrac{1}{\beta^2}\right)$ | $\mathcal{O}\left(\log\left(\dfrac{1}{|\beta|}\right)\right)$ |
| Horseshoe + | $\mathcal{O}\left(\dfrac{\log|\beta|}{\beta^2}\right)$ | $\mathcal{O}\left(\log^2\left(\dfrac{1}{|\beta|}\right)\right)$ |
| Dirichlet–Laplace | $\mathcal{O}\left(\dfrac{|\beta|^{a^*/2-3/4}}{\exp\{\sqrt{2|\beta|}\}}\right)$ | $\mathcal{O}\left(\dfrac{1}{|\beta|^{1-a^*}}\right)$ |
| Generalized Double Pareto | $\mathcal{O}\left(\dfrac{1}{|\beta|^{1+\alpha}}\right)$ | $\mathcal{O}(1)$ |
| R2D2 | $\mathcal{O}\left(\dfrac{1}{|\beta|^{1+2b}}\right)$ | $\mathcal{O}\left(\dfrac{1}{|\beta|^{1-2a_\pi}}\right)$ |

## H  EVALUATION METRICS

We summarize the evaluation metrics used in the experiments as follows.

- Accuracy: the fraction of correct predictions to the total number of ground truth labels.
- F-1 score: The F-1 score for each class is defined as

$$\text{F-1 score} = 2 \cdot \frac{\text{precision} \cdot \text{recall}}{\text{precision} + \text{recall}}$$

  where 'recall' is the fraction of correct predictions to the total number of ground truths in each class and precision is the fraction of correct predictions to the total number of predictions in each class.
- AUROC: the area under the receiver operating curve (ROC) which is the plot of the true positive rate (TPR/Recall) against the false positive rate (FPR).
- AUPR: the area under the precision-recall curve.

