# OpenReview forum: "R2D2-Net: Shrinking Bayesian Neural Networks via R2D2 Prior"
_ICLR.cc/2024/Conference — ICLR 2024 Conference Withdrawn Submission_

### Official Review · Reviewer_kYjv · 2023-10-31

**Soundness:** 1 poor
**Presentation:** 2 fair
**Contribution:** 2 fair
**Rating:** 3
**Confidence:** 3

**Summary:**

The paper proposes to apply the R2D2 prior, a global-local shrinkage prior, to Bayesian neural network parameters. The main motivation for using this prior is to obtain high concentration at zero and heavy tails, shrinking irrelevant coefficients effectively while preserving important features.  The paper also proposes a variational Gibbs inference algorithm to approximate weight posterior distribution. The authors demonstrate the effectiveness of this prior and inference scheme through experiments on synthetic regression datasets and image classification datasets.

**Strengths:**

- The work is well-motivated and the code is anonymously available.
- The idea of applying the R2D2 prior to Bayesian neural networks is novel.
- This paper tackles an important task in Bayesian deep learning, which is to design a good prior that promotes sparsity in neural networks while enhancing predictive performance.
- The paper is improved compared to the NeurIPS submission such as adding more related works on sparsity-inducing priors, and adding experiments with ResNets, but does not fully resolve all the concerns.

**Weaknesses:**

- The technical contribution is quite limited as the R2D2 Shrinkage prior is already proposed and the variational Gibbs inference scheme is widely used in the literature. For such a contribution, the paper should show strong empirical results. However, as mentioned, this version is improved compared to the NeurIPS submission by adding experiments with ResNets on CIFAR10. However, why is the performance of ResNets not very good as accuracies obtained by all the methods are less than 80%, as shown in Table 5? These are much lower than the existing results.
- In Table 5, why are the results of all the baselines much better than those in the NeurIPS submission?  From this and the previous bullet, the Reviewer has some doubts about the results and how the experiments were designed and conducted. In the previous results, there were huge gaps between the proposed method and the baselines. In this version, they are somehow comparable. In addition, the authors should show the standard deviation obtained from multiple runs or conduct significance tests for this table, as the performance differences between methods are very small.
- The choice of baselines seems inadequate, as the authors only consider the Horseshoe prior as the main baseline. The authors should compare to other sparsity-inducing priors such as log-uniform, Group horseshoe prior, or spike-and-slab priors [1,2,3] .
- The paper is not very well-written. For example, in section 4.1, it would greatly enhance the clarity and understanding of the work if the authors were to provide intuitions regarding how the R2D2 prior facilitates network sparsification.

References

[1] Louizos et al. Bayesian Compression for Deep Learning. NIPS 2017

[2] Molchanov et al. Variational Dropout Sparsifies Deep Neural Networks. ICML 2017

[3] Bai et al. Efficient Variational Inference for Sparse Deep Learning with Theoretical Guarantee. NeurIPS 2020.

**Questions:**

- In the paragraph “Sparsifying Neural Networks” of Section 2, the authors claim that “Despite the similarity in approaches, our work focuses on a design on BNN with shrinkage priors which can improve its capability (i.e., predictive and uncertainty estimation performance) instead of compressing the existing architecture.” Why does shrinking neural networks improve predictive and uncertainty performance? Could the authors elaborate more on this?

- Throughout the paper, the authors claim that the proposed approach yields the smallest predictive variance.  Why is a smaller predictive variance considered advantageous? To evaluate the uncertainty, the authors should use metrics like log-likelihood, ECE. For the experiment on the synthetic regression datasets, it would be great if the author could evaluate the proposed method on “in-between” and out-of-domain data.

---

### Official Review · Reviewer_JJfw · 2023-11-01

**Soundness:** 2 fair
**Presentation:** 2 fair
**Contribution:** 2 fair
**Rating:** 3
**Confidence:** 3

**Summary:**

This paper proposed R2D2-Net, a Bayesian neural network model with $R^2$-induced Dirichlet Decomposition prior. A variational Gibbs inference method is used to learn the model. Experiments show that the R2D2-Net has better predictive performance and shrinkage performance.

**Strengths:**

Using a Bayesian neural network model with a shrinkage prior to compress the model and select important features is an interesting topic. The paper presents extensive numerical experiment to support the superior performance.

**Weaknesses:**

1. The R2D2 prior and the Gibbs inference algorithm(with slight modification) are from previous work [1], which makes the contribution of the paper less significant. The author discussed the difference in Appendix A. From what I understand, the main contribution is to apply the R2D2 prior to DNN setting and conduct extensive experiments to verify results.

2. For the experiments, I appreciate the authors for including various experiments, but I think the significance of the results are not very well presented:
(1)  For prediction performance, I am not sure why the accuracies of AlexNet on CIFAR and TinyImageNet are so low. And in section 6.2, it says "It is noteworthy that since BNNs impose a natural regularization on the weights, it is difficult for BNN designs to outperform their frequentist counterpart." I agree that with parameter compression, the model can perform worse, but this also makes the model less useful. For a fair comparison, maybe it can follow the network pruning literature by comparing the performance of different methods with similar pruning ratios. Otherwise, considering the regularization effect, it is hard to say if the comparison of predictive performance to other shrinkage priors is fair.
(2) For Shrinkage Performance, the paper presents a parameter density plot around 0 and convolutional filters of different networks. But why having parameters more concentrated around 0 useful, or why in Figure 5, R2D2 filters are more meaningful(e.g. what can we say about the meaning of the filters in Figure 5)? I think more concrete experiments can be variable selection on synthetic data sets, see if the model can select correct variables, or network pruning experiments, maybe having weights more concentrated around 0 can lead to a better pruning method(i.e. removing parameters have less effect), this would help reduce the network size.

3. The paper emphasizes that the R2D2 prior possesses the largest concentration rate at zero and the heaviest tail. From the above point 2, I think the paper didn't present the advantage of having those properties in the experiments very well.  Perhaps these are related to theoretical properties such as posterior concentration, e.g. [2][3]. Maybe more discussion on why these properties are desirable will be helpful.


### Reference
[1] Zhang, Yan Dora, et al. "Bayesian regression using a prior on the model fit: The r2-d2 shrinkage prior." Journal of the American Statistical Association 117.538 (2022): 862-874.

[2] Polson, Nicholas G., and Veronika Ročková. "Posterior concentration for sparse deep learning." Advances in Neural Information Processing Systems 31 (2018).

[3] Sun, Yan, Qifan Song, and Faming Liang. "Consistent sparse deep learning: Theory and computation." Journal of the American Statistical Association 117.540 (2022): 1981-1995.

**Questions:**

1. I think spike-and-slab type priors are also frequently used sparse inducing priors. The references [2] [3] given in the weakness apply those priors to Bayesian neural networks as well. Can the authors comment a bit on the relationship of these priors compared to R2D2 priors?

---

### Official Review · Reviewer_XdB3 · 2023-11-06

**Soundness:** 3 good
**Presentation:** 3 good
**Contribution:** 2 fair
**Rating:** 6
**Confidence:** 3

**Summary:**

The authors propose using the R2-induced Dirichlet decomposition shrinkage prior for Bayesian neural networks. The authors empirically demonstrate that the proposed prior can effectively shrink coefficients associated to irrelevant features towards zero, and vice versa for relevant features. Moreover, the authors also propose using variational Gibbs inference to approximate the posterior distribution of weights and use closed-form expressions for the KL divergence computation. Empirical results show that the obtained BNNs are competitive with the baselines that were considered in this work.

**Strengths:**

The axes of strength of this paper are the following:

- Clarity: The motivation is clearly stated, the related work section is extensive, and all key notions are properly introduced. The paper is overall easy to follow. I also like figure 1 as it summarizes the method very clearly.
- Soundness: The proposed pipeline, including the prior, the variational Gibbs inference, and the KL divergence estimation, is theoretically sound.
- Evaluation: I find the evaluation to be quite extensive, and appreciate including the frequentist and the ensembling performances in the tables as well. I would have hoped to see the authors consider the best performing model for all these datasets instead, which can definitely achieves much better accuracy on the considered dataset.

**Weaknesses:**

The originality and significance of this work are the biggest weakness of this work. The paper combines 3 components that already exist in the literature: shrinkage priors, variational Gibbs inference, and estimating the KL divergence with variational posterior distributions. The contribution of this work is very incremental and marginal in my opinion. Maybe the authors can consider the covariate shift scenario (see questions below) to further expand the significance of their work.

**Questions:**

- It has been shown that Gaussian priors fail to generalize well under covariate shift [1]. How would this new prior perform under covariate shift in the same setup? Including these covariate shift scenarios might actually increase the significance of this work.
- How can Deep Ensemble perform worse than the frequentist method -- not even within the variances range? That is quite strange to me.

[1] Izmailov, P., Nicholson, P., Lotfi, S. and Wilson, A.G., 2021. Dangers of Bayesian model averaging under covariate shift. Advances in Neural Information Processing Systems, 34, pp.3309-3322.